# Investigation of Growth Factors and Mathematical Modeling of Nutrient Media for the Shoots Multiplication In Vitro of Rare Plants of the Rostov Region

**Vasiliy A. Chokheli** [1,*], **Semyon D. Bakulin** [2], **Olga Yu. Ermolaeva** [1], **Boris L. Kozlovsky** [1], **Pavel A. Dmitriev** [1], **Victoriya V. Stepanenko** [1], **Igor V. Kornienko** [1,3], **Anastasia A. Bushkova** [1], **Vishnu D. Rajput** [1] and **Tatiana V. Varduny** [1]

1. Academy of Biology and Biotechnology, Southern Federal University, Rostov on Don 344090, Russia
2. Department of Botany, Moscow Timiryazev Agricultural Academy, Russian State Agrarian University, Plant Breeding and Seed Technology, Moscow 127550, Russia
3. Southern Scientific Center of the Russian Academy of Sciences, Paleogeography Laboratory, Rostov-on-Don 344006, Russia
* Correspondence: vachokheli@sfedu.ru

**Abstract:** Micropropagation is an effective way to preserve the gene pool of threatened plants. This study is devoted to the mathematical modeling of nutrient media and the study of the effect of *m*T (meta-topoline) on the multiplication of shoots of *Hedysarum grandiflorum*, *Hyssopus cretaceus*, and *Matthiola fragrans* in vitro in comparison with benzylaminopurine (BAP) and kinetin (KT). Initiation was performed on an MS medium with 0.5 mg/L BAP. For shoots multiplication, MS, B5, and WPM media were used with the addition of *m*T, BAP, KT. For *H. grandiflorum*, the multiplication coefficient of shoots was highest on medium B5 with the addition of mT at a concentration of 1 mg/L—2.90 shoots per plant, for *H. cretaceus*—B5 + 0.5 mg/L *m*T, and for *M. fragrans*—B5 + 1 mg/L KT. A positive effect of *m*T on *H. grandiflorum* and *M. fragrans* in vitro was found. The efficiency of using KT for *H. cretaceus* shoot multiplication is shown. The effectiveness of the B5 nutrient medium for *H. grandiflorum* and *M. fragrans* was determined. The positive effect of WPM for *H. cretaceus* micropropagation has been demonstrated. It is not recommended to use the MS media for micropropagation of these plant species.

**Keywords:** red list; 6-benzylaminopurine; meta-topoline; kinetin; micropropagation; phytohormones

## 1. Introduction

Nowadays, one of the most popular and highly effective methods of preserving the gene pool of rare and endangered plant species is micropropagation. This method not only make it possible to preserve endangered plant species in vitro but also to study their genetic, physiological, anatomical, and morphological aspects of biology, to identify and isolate secondary metabolites that are used in medicine, as well as to produce an amount of material sufficient for breeding or reproduction of regenerating plants for further sale [1–3]. The improvement of old and the search for new techniques is necessary due to introducing new plant species into in vitro conditions. The selection of the optimal phytohormonal composition of nutrient media can contribute not only to an increase in the degree of multiplication but also to the effective production of callus, and the creation of bioreactor cultures, tissue cultures, embryos, and pollen [3].

The flora of the Rostov region is rich in rare and endangered plant species. Nowadays, more than 273 species of plants and fungi with a threatened status have been included in the Red List of the region, and 45 plant species from this list are included in the Red List of the Russian Federation [4]. For threatened flora species of the Rostov region, such as *Hedysarum grandiflorum* Pall. and *Hyssopus cretaceus* Dubj., data on the methods of

microproporation vary, and no microclonal propagation schemes have been developed for *Matthiola fragrans* Bunge.

The studies on the microproporation of these and related species recommend use of benzylaminopurine (BAP) as the only cytokinin [5,6]. Growth regulators such as kinetin (KT) [7], 2-isopentyladenine [8], and tidiazuron [9], are rarely used. Many plants, especially threatened ones, are characterized by difficulties in in vitro cultivation. Such phenomena may be caused by seed dormancy, difficulties in sterilizing explants, low rates of shoot multiplication and rhizogenesis, and problems with histogenesis and morphogenesis of plants [10]. The success of in vitro cultivation of plants and their further acclimatization strongly depends on the choice of a suitable phytohormonal growth regulator. It is known that BAP in plant tissues is metabolized to stable toxic compounds—6-benzylaminopurine-9-glycosides (9-B-glucopyransosyl-benzyladenine), which accumulate in tissues at the base of plants, inhibiting their further development [11].

An alternative to BAP is N6-(3-hydroxybenzyl)adenine (meta-topoline, *m*T), a less toxic phytohormone that improves further processes of root formation in vitro and acclimatization. The *m*T strengthens the processes of multiplication and rhizogenesis, non-irrigation of tissues, and increases the success of acclimatization of plants [12,13]. Successful cultivation of plants in tissue culture depends not only on the choice of growth stimulants and their concentration, but also depends on the composition of the components of the medium. In this regard, the necessary stage is the mathematical planning of the nutrient medium [14].

The purpose of this research is to mathematically plan and develop effective methods of multiplication of plant species in vitro to determine the influence of BAP and KT on the *m*T growth and development of plants.

## 2. Materials and Methods

*Hedysarum grandiflorum* is an herbaceous perennial with ornamental, forage, and medicinal properties [4,15]. *Hedysarum cretaceus* is a perennial herb, and an obligate inhabitant of chalk outcrops. It has decorative, medicinal, ethiromalenic, and honey-bearing properties [4,16]. *M. fragrans* is an herbaceous perennial inhabitant of chalk outcrops, and an ornamental plant [4]. In the Red List of the Rostov region, *H. grandiflorum* and *H. cretaceus* have the status of 3 b, d—rare species with narrow ecological confinement associated with specific growing conditions, having a limited range, part of which is located in the Rostov region. *Matthiola fragrans* has the status of 3 b—a rare species with a narrow ecological amplitude associated with a specific substrate for growth. These species are listed in the Red Book of the Russian Federation under the status 3—rare species [4]. They experience a strong negative impact from the destruction of natural habitats, low competitiveness, and anthropogenic disturbances of the habitat. All three species are plants that prefer chalky soil substrates [4,17].

Seeds collected at the Nursery of Rare and Endangered Plants of the Botanical Garden of the Southern federal university (BG SFedU) were used as explants for the introduction of research objects into the culture.

For *H. cretaceous* and *M. fragrans*, the seed material was sterilized by washing the seeds in a mixture of 70% ethyl alcohol solution and 3% hydrogen peroxide solution in a ratio of 1:1 for 10 min, followed by washing the diaspores in distilled water for 10 min four times. *Hedysarum grandiflorum* seeds, after washing in a mixture of alcohol and hydrogen peroxide, were immersed in 96% ethanol for 1 s, and then burned with a gas burner flame. This is necessary to get rid of the hard seed. The seeds were germinated on a Murashige-Skoog nutrient medium with the addition of BAP at a concentration of 0.5 mg/L [18].

When developing a nutrient medium for the cultivation of rare plants, it is necessary to carry out mathematical planning of the main components (factor gradations); this is the type of phytohormone, its concentration, and mineral-organic base (nutrient medium). The course of planning is based on the following algorithm: (1) selection of 3 or more different bases for nutrient media (MS, B5, etc.), including fractional bases (1/2 MS, 1/3 MS, etc.); (2) selection of phytohormone (BAP, *m*T, etc.); (3) selection of phytohormone concentration

in large steps (0.5 mg/L): 0 (hormone-free control medium), 0.5 mg/L, 1 mg/L, 1.5 mg/L, 2 mg/L. If necessary, you can take large concentrations. This is what concerns the animation of escapes. For rhizogenesis, it is recommended to use the following concentrations of phytohormones (in step of 0.2 mg/L): 0 (hormone-free control medium), 0.2 mg/L, 0.4 mg/L, 0.6 mg/L, 0.8 mg/L, 1 mg/L. If necessary, you can also take large concentrations. After setting up all the experiments, the data is calculated by multivariate analysis of variance (ANOVA) and the influence of the factor is determined.

The second part of the experimental development on nutrient media planning is to reduce the step (0.1 mg/L) of the phytohormone concentration on a certain nutrient medium. For example, if the effective basis turned out to be a WPM medium with a phytohormone *m*T concentration equal to 1.5 mg/L, then it is worth reducing the step in one direction (1.4 mg/L, 1.3 mg/L) and increasing the step in the other direction (1.6 mg/L, 1.7 mg/L). As a result, the optimal nutrient medium for this genotype of a rare plant is obtained.

Our study presents the first part of the experiment and shows the influence of various factors on 3 species from 3 genera belonging to 3 different families: *Fabaceae* family (*Hedysarum grandiflorum*), Lamiaceae family (*Hyssopus cretaceus*), and the Brassicaceae family (*Matthiola fragrans*).

MS, Gamborg and Eveleg media, Woody Plant Medium [19], with the adding of several phytohormones in different concentrations, were used to stimulate the multiplication of shoots (Table 1).

**Table 1.** Variants of experimental media for stimulating the animation of shoots of research objects *in vitro*.

|  | **MS** | **B5** | **WPM** |
|---|---|---|---|
| Control | - | - | - |
| BAP | 0.5<br>1.0<br>1.5<br>2.0 | 0.5<br>1.0<br>1.5<br>2.0 | 0.5<br>1.0<br>1.5<br>2.0 |
| *m*T | 0.5<br>1.0<br>1.5<br>2.0 | 0.5<br>1.0<br>1.5<br>2.0 | 0.5<br>1.0<br>1.5<br>2.0 |
| KT | 0.5<br>1.0<br>1.5<br>2.0 | 0.5<br>1.0<br>1.5<br>2.0 | 0.5<br>1.0<br>1.5<br>2.0 |

Plant passages were carried out in a laminar box using sterilized instruments (scissors, tweezers, dissecting needles). The pH of the nutrient media was adjusted to a value of 6.0 using a 1 M KOH solution. The culture media was sterilized in an autoclave MLS-3751L (Sanyo) at a temperature of 121 °C and pressure of 1.5 atmospheres for 30 min. The plants were cultivated at a constant temperature of 25 °C and a 16-h photoperiod.

The sample for each variant of the experiment was 20 plants for each media combination. The main parameter to assess the success of the media combinations revealed in the course of the study was the multiplication coefficient of the shoots. The quadratic error was determined for the obtained parameter values, and a comparative analysis was performed using the Student's *t*-test at $p = 0.05$ [20]. Statistical analysis of the values of the proliferation coefficient was carried out using the method of multivariate analysis of variance ANOVA using programs Statistica 13.3 and Microsoft Excel.

## 3. Results

### 3.1. Sterilization

The results of the use of the described scheme of sterilization of seeds of *H. grandiflorum*, *H. cretaceus*, and *M. fragrans* are shown in the diagram (Figure 1).

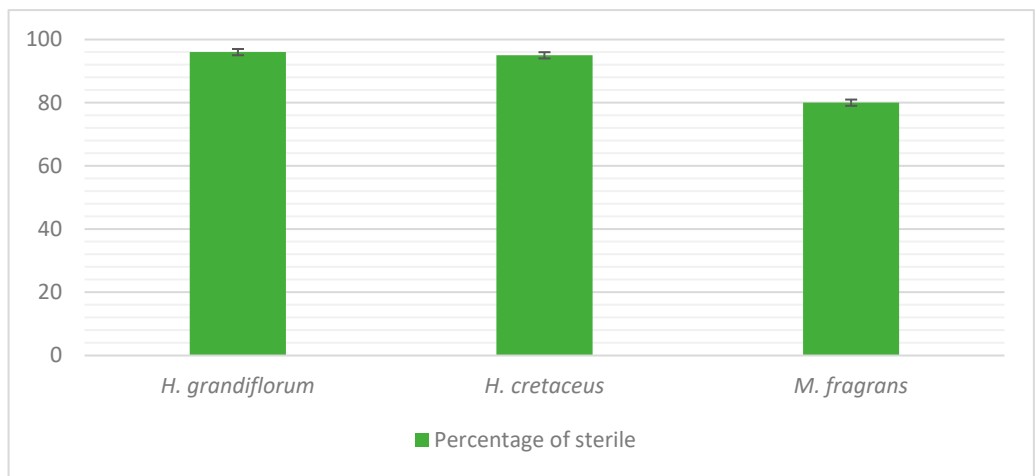

**Figure 1.** Results of application of methods of sterilization of seeds of objects of research.

It was managed to achieve, respectively, (mean $\pm$ standard error), 96.00 $\pm$ 2.77%, 95.00 $\pm$ 3.08%, and 80.00 $\pm$ 5.65% of the seed sterility level.

### 3.2. Multiplication

The average values of the multiplication coefficient of shoots of the studied plant species on different nutrient media were revealed. Standard errors for the obtained values are determined (Tables 2–4).

**Table 2.** Average values of *H. grandiflorum* multiplication coefficient on different nutrient media.

|  |  | MS | B5 | WPM |
|---|---|---|---|---|
| Control |  | 1.00 $\pm$ 0.00 * | 1.00 $\pm$ 0,00 * | 1.7 $\pm$ 0.22 * |
| *m*T | 0.5 | 1.00 $\pm$ 0.00 * | 2.10 $\pm$ 0.16 * | 1.7 $\pm$ 0.23 * |
|  | 1.0 | 1.00 $\pm$ 0.00 * | 2.90 $\pm$ 0.34 | 1.3 $\pm$ 0.11* |
|  | 1.5 | 1.00 $\pm$ 0.00 * | 2.40 $\pm$ 0.15 | 2.00 $\pm$ 0.21* |
|  | 2.0 | 1.00 $\pm$ 0.00 * | 2.45 $\pm$ 0.23 | 1.30 $\pm$ 0.13* |
| BAP | 0.5 | 1.75 $\pm$ 0.19 * | 1.25 $\pm$ 0.01 * | 1.60 $\pm$ 0.18 * |
|  | 1.0 | 2.35 $\pm$ 0.26 | 1.60 $\pm$ 0.11 * | 1.20 $\pm$ 0.09 * |
|  | 1.5 | 1.20 $\pm$ 0.14 * | 1.30 $\pm$ 0.11 * | 1.65 $\pm$ 0.20 * |
|  | 2.0 | 1.55 $\pm$ 0.14 * | 1.70 $\pm$ 0.19 * | 1.35 $\pm$ 0.15 * |
| KT | 0.5 | 1.35 $\pm$ 0.14 * | 1.55 $\pm$ 0.14 * | 1.25 $\pm$ 0.10 * |
|  | 1.0 | 1.10 $\pm$ 0.07 * | 1.90 $\pm$ 0.1 * | 1.20 $\pm$ 0.09 * |
|  | 1.5 | 1.15 $\pm$ 0.08 * | 1.55 $\pm$ 0.14 * | 1.35 $\pm$ 0.13 * |
|  | 2.0 | 1.40 $\pm$ 0.13 * | 2.65 $\pm$ 0.23 | 1.35 $\pm$ 0.11 * |

* The values with a significant statistical difference from the highest value of the parameter according to the Student's t-criterion at $t_{teor}$ = 2.09, $\alpha$ = 5%, $p$ = 0.05 are emphasized.

**Table 3.** Average values of *H. cretaceus* multiplication coefficient on different nutrient media.

| | | MS | B5 | WPM |
|---|---|---|---|---|
| Control | | 1.00 ± 0.00 * | 1.45 ± 0.18 * | 1.80 ± 0.25 * |
| *m*T | 0.5 | 1.00 ± 0.00 * | 4.20 ± 0.42 * | 3.00 ± 0.32 * |
| | 1.0 | 1.15 ± 0.08 * | 2.05 ± 0.20 * | 3.00 ± 0.36 * |
| | 1.5 | 1.60 ± 0.23 * | 1.90 ± 0.20 * | 2.25 ± 0.19 * |
| | 2.0 | 1.20 ± 0.16 * | 1.85 ± 0.20 * | 3.50 ± 0.52 |
| BAP | 0.5 | 1.40 ± 0.15 * | 1.60 ± 0.20 * | 3.20 ± 0.46 |
| | 1.0 | 2.60 ± 0.22 * | 1.75 ± 0.20 * | 1.95 ± 0.22 * |
| | 1.5 | 2.20 ± 0.21 * | 1.30 ± 0.15 * | 1.45 ± 0.15 * |
| | 2.0 | 2.40 ± 0.23 * | 1.65 ± 0.21 * | 1.00 ± 0.00 * |
| KT | 0.5 | 2.25 ± 0.19 * | 2.60 ± 0.20 * | 2.00 ± 0.24 * |
| | 1.0 | 2.10 ± 0.20 * | 1.85 ± 0.23 * | 2.30 ± 0.24 * |
| | 1.5 | 2.85 ± 0.33 * | 2.25 ± 0.25 * | 2.45 ± 0.26 * |
| | 2.0 | 2.35 ± 0.31 * | 1.80 ± 0.20 * | 2.60 ± 0.22 * |

* The values with a significant statistical difference from the highest value of the parameter according to the Student's t-criterion at $t_{teor}$ = 2.09, $\alpha$ = 5%, $p$ = 0.05 are emphasized.

**Table 4.** Average values of *M. fragrans* multiplication coefficient on different nutrient media.

| | | MS | B5 | WPM |
|---|---|---|---|---|
| Control | | 1.95 ± 0.14 * | 2.90 ± 0.28 | 3.05 ± 0.22 |
| *m*T | 0.5 | 2.65 ± 0.27 * | 3.25 ± 0.36 | 3.15 ± 0.30 * |
| | 1.0 | 2.75 ± 0.37 | 1.95 ± 0.22 * | 2.55 ± 0.23 * |
| | 1.5 | 2.85 ± 0.34 | 1.95 ± 0.22 * | 2.10 ± 0.22 |
| | 2.0 | 3.05 ± 0.38 | 2.40 ± 0.26 * | 3.35 ± 0.39 * |
| BAP | 0.5 | 1.00 * | 2.90 ± 0.45 * | 1.40 ± 0.18 * |
| | 1.0 | 1.00 * | 2.40 ± 0.37 * | 1.75 ± 0.22 * |
| | 1.5 | 1.00 * | 2.35 ± 0.27 * | 1.75 ± 0.22 * |
| | 2.0 | 1.00 * | 2.30 ± 0.33 * | 1.85 ± 0.24 * |
| KT | 0.5 | 1.85 ± 0.20 * | 1.65 ± 0.20 * | 2.65 ± 0.25 * |
| | 1.0 | 1.90 ± 0.14 * | 3.75 ± 0.40 * | 1.90 ± 0.27 * |
| | 1.5 | 1.80 ± 0.29 * | 2.30 ± 0.24 * | 2.65 ± 0.30 * |
| | 2.0 | 2.60 ± 0.36 * | 2.75 ± 0.33 * | 2.40 ± 0.24 * |

* The values with a significant statistical difference from the highest value of the parameter according to the Student's t-criterion at $t_{teor}$ = 2.09, $\alpha$ = 5%, $p$ = 0.05 are emphasized.

The highest average value of the multiplication coefficient of *H. grandiflorum* shoots *in vitro* was found on nutrient medium B5 + 1 mg/L *m*T—2.90 ± 0.34 shoots per plant (Table 2, Figure 2). A comparative analysis using the *t*-test indicates the reliability of this result compared to most parameter values when using other nutrient media options. The differences with the use of 1 mg/L of BAP on MS medium and 2 mg/L of KT on mineral base B5 are unreliable. The range of the multiplication coefficient when using *m*T varied from 1.00 to 2.90 ± 0.34 shoots per plant. The use of BAP resulted in obtaining average values of the multiplication coefficient in the range from 1.20 ± 0.14 to 2.35 ± 0.26 shoots per plant. The variation of the values of the analyzed parameter on media with KT ranged from 1.10 ± 0.07 to 2.65 ± 0.23. It was when using the B5 nutrient medium that both *m*T and other growth regulators used proved to be more effective than on other mineral bases. When using the MS mineral base, the multiplication coefficient reached its maximum in the variant with the addition of 1 mg/L of BAP—2.35 ± 0.26 per plant. The maximum value of the multiplication coefficient on the B5 medium was recorded using *m*T at a concentration of 1 mg/L—2.90 ± 0.34 shoots per plant. At the same time, the coefficient values are distributed more evenly on the WPM medium than in variants with other mineral bases.

The highest average value of the multiplication coefficient of *H. cretaceus* shoots in vitro was found on nutrient medium B5 + 0.5 mg/L *m*T—4.20 ± 0.42 shoots per plant (Table 3, Figure 3). A comparative analysis using the *t*-test indicates the reliability of

this result compared to most parameter values when using other nutrient media options. The differences with the use of 2 mg/L *m*T and 0.5 mg/L BAP on a mineral basis WPM are unreliable. Variants of experiments with *m*T showed relatively high values of the multiplication coefficient on B5 and WPM media (Table 3). BAP demonstrates a less stimulating effect on all media (from 1.00 to 3.20 ± 0.46 shoots per plant). The maximum value of the multiplication coefficient when using BAP is observed at its concentration of 0.5 mg/L in the WPM medium. When using KT as a growth regulator, the multiplication coefficient varied less than when using mT and BAP—from 1.80 ± 0.20 to 2.85 ± 0.33 shoots per plant. There is a linear increase in the multiplication coefficient when using KT in a WPM environment.

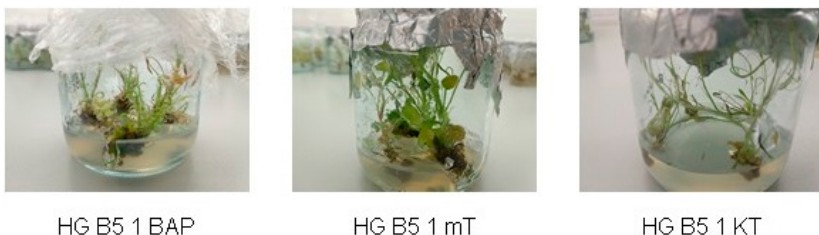

**Figure 2.** The effect of different types of growth regulators on plants of *H. grandiflorum* shoots *in vitro*.

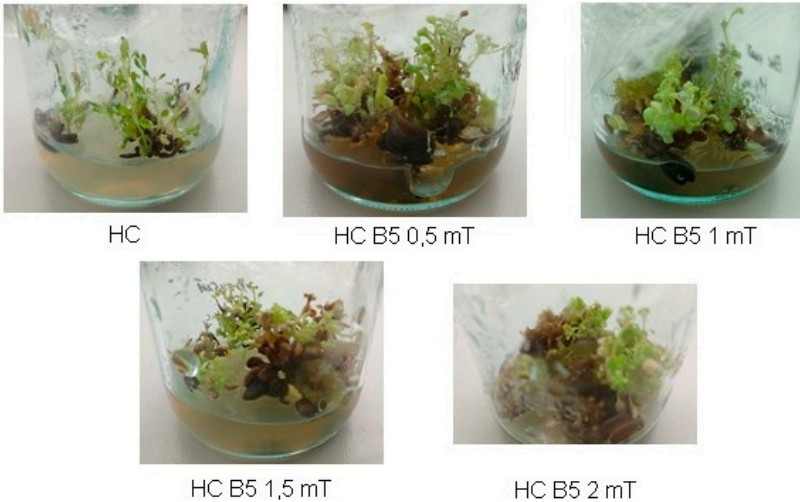

**Figure 3.** The effect of different concentrations of *m*T on plants of *H. cretaceus* shoots *in vitro*.

The highest average value of the multiplication coefficient of *M. fragrans* shoots in vitro was found on nutrient medium B5 + 1 mg/L KT (Figure 4). On all variants of nutrient media with the addition of mT, relatively high values of the multiplication coefficient are observed—from 1.95 ± 0.22 to 3.35 ± 0.39 shoots per plant (Table 4). The maximum value is fixed at WPM + 2 mg/L mT. Significant differences were revealed in most cases of comparison with media using BAP, where the spread of results varied from 1.00 to 2.90 ± 0.45 shoots per plant. At the same time, the maximum value of the parameter is fixed at the lowest concentration of BAP on medium B5—0.5 mg/L. When using this mineral base, relatively high values with BAP were observed, although they did not give significant differences when compared with most other experimental options. The multiplication coefficient on nutrient media with the addition of KT varied from 1.65 ± 0.20 (B5 + 0.5 mg/L KT) to 3.75 ± 0.40 (B5 + 1 mg/L KT). The last value of the coefficient turned out to be the maximum for the entire experiment. Based on comparative analysis, the use of KT at a concentration of 1 mg/L on B5 medium does not differ significantly from the use of mT at most concentrations, as well as BAP at a concentration of 0.5 mg/L on all types of mineral bases studied.

When using KT as a growth regulator, regardless of the type of mineral base, hormone, and concentration, the plants looked green, devoid of signs of chlorosis, vitrification, and

necrosis. When using BAP, vitrification of plants was observed. Such growth features were recorded more intensively when using the MS mineral base, both together with BAP and *m*T, but to a lesser extent, at a concentration of *m*T 2 mg/L.

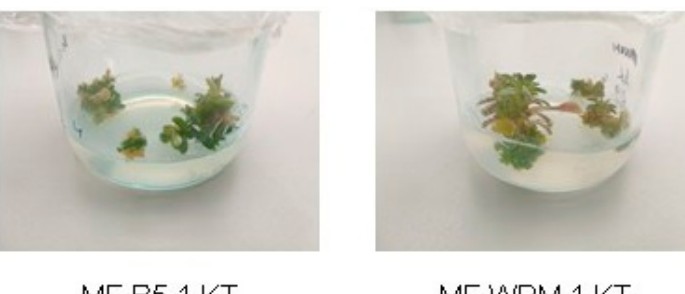

MF B5 1 KT      MF WPM 1 KT

**Figure 4.** The effect of different media with *m*T on plants of *H. cretaceus* shoots *in vitro*.

### 3.3. Statistical Analysis

Statistical comparisons were made between all variants of nutrient media for each studied species using the Student's *t*-test at *p* = 0.05 (Tables 2–4).

Multivariate analysis of the values of the multiplication coefficient was performed. The factors were: mineral base type (MS, B5, WPM), growth regulator (*m*T, BAP, KT), and growth regulator concentration, mg/L (0.0; 0.5; 1.0; 1.5; 2.0). Numerical results of the analysis are presented in Appendix A Table A1, Table A2, Table A3, and graphical data are shown in Figure 5.

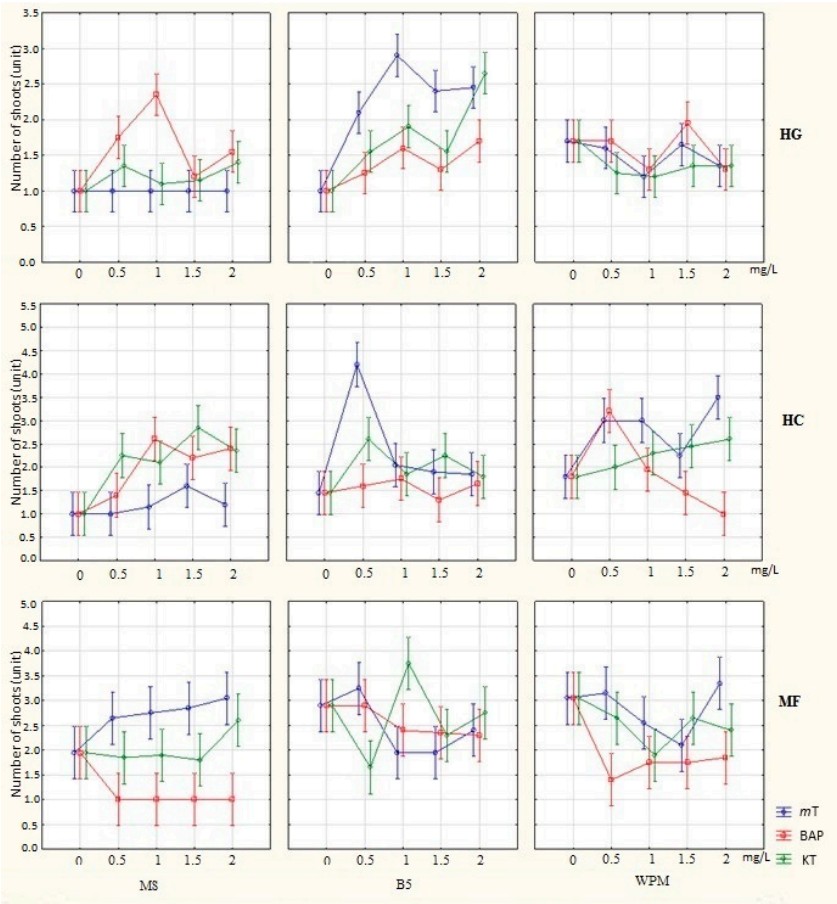

**Figure 5.** The combined effect of mineral base, growth regulator, and growth regulator concentration on the average number of *H. grandiflorum* (HG), *H. cretaceous* (HC), and *M. fragrans* (MF) shoots *in vitro*. Confidence interval = 0.95.

## 4. Discussion

### 4.1. Seed Sterilization

Roasting in the flame had a scarifying effect on the seeds of *H. grandiflorum*, which have a hard-seed surface. The firing efficiency was also shown earlier by us using the example of *H. cretaceum* seeds [21]. The results of sterilization of *H. grandiflorum* seeds were higher compared to the results given in some other studies on the micropropagation of *H. grandiflorum* and other species of the genus, where diacid, 70% alcohol solution, sodium hypochlorite, and Domestos were used as sterilizing agents [6,22,23].

Sterilization of *H. cretaceus* seeds according to the described scheme allowed the obtainment of 95% sterile seeds. This methodic sterilization of primary explants was successfully tested by us earlier on seeds of another species—*H. angustifolius* [3]. The chosen technique is not inferior in its effectiveness to other methods of sterilization of seeds of this species using sodium hypochlorite [9,24], silver nitrate [25], "Lysoformin 3000" [26], 70% ethanol [9,26], mercury chloride [27], chloramine B [25], and Tween 20 [24].

The result of using the chosen sterilization technique makes it more effective for *M. fragrans* seeds than using sodium hypochlorite, as indicated in the literature [28]. To achieve a higher percentage of sterile seeds, it may be necessary to slightly reduce the exposure time of seeds in sterilizers.

### 4.2. Multiplication of Shoots

#### 4.2.1. Hedysarum Grandiflorum

Interestingly, for the micropropagation of other members of the genus *Hedysarum*, it is recommended to use BAP, sometimes KT, as suitable cytokinins in a fairly wide range—from 0.1 to 10.0 mg/L, often together with various auxins [5,29–33]. For several representatives of the genus *Hedysarum* (*H. cumuschtanicum, H. cephalotes, H. songoricum, H. ferganense, H. neglectum, H. semenovii, H. sultanovae, H. montanum, H. plumosum*), it is proposed to use the nutrient medium of Gamborg and Eveleg (B5) with the addition of BCI and BAP in concentrations of 0.25 mg/L and 0.1 mg/L, respectively, as a substrate for multiplication of shoots. Variants of the B5 nutrient medium with the addition of 0.1 mg/L of KT and BAP separately also proved to be effective. Medium B5 with the addition of 0.25 mg/L of BCI had the greatest stimulating effect on the process of rhizogenesis of regenerating plants of all studied plants [34].

Visually, the healthiest, developed *H. grandiflorum* plants looked on medium B5 using *m*T and KT as growth regulators. The *m*T effectively stimulates multiplication, and KT—elongation of shoots. At the same time, the use of nutrient media with the addition of BAP causes vitrification of plants when using all mineral bases. With an increase in the concentration of BAP, the degree of vitrification increased. When using the mineral base of WPM, the plants looked weak, and chlorosis was often detected. This may be caused by an insufficient amount of nitrogen in the composition of the medium compared with other variants of mineral bases [18,19,35].

MS mineral base is known for its high content of both nitrate and ammonium nitrogen. We assume that ammonium nitrogen in high concentrations can block the proliferation and multiplication of *H. grandiflorum* shoots *in vitro*. There are known data on the effect of the ratio of nitrate and ammonium on the organogenesis of various plants *in vitro*. Thus, Bennett et al. [36] found that with the complete removal of ammonium nitrate from the nutrient medium, the rooting efficiency and survival rate during the adaptation of *Eucalyptus globulus* plants increase. A decrease in the concentration of ammonium nitrogen compared to nitrate increased the fresh weight of potato plants *in vitro*, made the pH level of the medium more stable, but did not increase the efficiency of plant multiplication [37]. An increase in the concentration of nitrate–nitrogen to ammonium led to a better degree of micro-cloning of garlic plants [38].

Numerical results of the analysis of variance (Appendix A Table A1) show that the degree of multiplication of *H. grandiflorum* shoots in vitro slightly depends on the selected regulator. The type of mineral base and the concentration of growth regulators have a

greater effect on changing this parameter. It can be seen from the graphs in Figure 5 that the greatest variation in values is observed when using phytohormones at a concentration of 1 mg/L. More definite patterns of the dependence of the effectiveness of animation on the selected types of hormone (Figure 2) and the mineral basis of the nutrient medium can be traced when using growth regulators at concentrations of 0.0 and 2.0 mg/L. Transitional situations are observed due to the use of hormones in concentrations of 0.5 and 1.5 mg/L. Interestingly, in most variants of the experiment, the effect of phytohormones was more noticeable on the B5 medium and to a lesser extent when using MS and WPM mineral bases. Application of *m*T had no effect on the multiplication of shoots in combination with the mineral base of MS.

The data obtained are consistent with the literature information on the effectiveness of using the B5 medium or its components for microcloning of representatives of the Fabaceae family [5,39,40]. For the embryogenesis of *Albizzia lebbeck*, a modified nutrient medium B5 was used, with the addition of phytohormones 2,4-D and Kin at concentrations of 0.5 mg/L and 2 mg/L, respectively [39]. For in vitro germination of seeds of *Vigna subterranea* as a mineral base, medium B5 was used, along with media such as MS, SH, and CHU [40].

### 4.2.2. Hysoppus Cretaceus

Low concentrations of cytokinins, up to 1 mg/L, are indicated as effective in studies on micropropagation of *Hyssopus* representatives [3,8,9,41,42]. Despite the effectiveness of *m*T in stimulating multiplication, plants on all media with its addition acquired chlorosis, and from the middle of the third week of cultivation, tissue necrosis. A similar situation was observed on all media with BAP, which is why vitrification and stunting also developed in plants. The absence of such adverse effects was observed when using KT as growth regulators, which additionally stimulated the elongation of shoots of the studied plants. BAP is often recommended together with auxins for in vitro micropropagation of shoots of other members of the genus—*H. officinalis*, *H. angustifolius* [9,25].

The actual data of the dispersion analysis show that in the case of *H. cretaceus*, the type of mineral medium, the choice of the growth regulator, and its concentration have both individually and together an impact on the effectiveness of the multiplication of shoots (Appendix A Table A2). According to the graphs in Figure 5, a similar pattern of action on the animation of shoots is observed for all used growth regulators at concentrations of 1.0 and 1.5 mg/L (Figure 3). Here, *m*T is more effective in the B5 media, BAP, and KT—in the MS media. At the same time, both BAP and KT reduce their effectiveness of action at given concentrations on the B5 medium. When using mineral medium B5, there is an increase in the efficiency of the use of all phytohormones at a concentration of 0.5 mg/L, especially *m*T. On the WPM medium, at this concentration, the use of BAP shows itself best of all. At a concentration of growth regulators of 2 mg/L, a similar efficiency dynamic is observed between *m*T and KT, whereas BAP in the range from MS to WPM becomes less effective for multiplying shoots. A low concentration of cytokinins also has a more effective effect on *H. angustifolius* multiplication in vitro [3].

Unfortunately, the above scientific literature does not explain the detrimental effect of *meta*-topoline on the growth and development of *H. cretaceous* regenerating plants *in vitro*. Being a "mildly" acting cytokinin, KT in the case of *H. cretaceus* may become more suitable for the multiplication of shoots in vitro, which has been tested by other researchers on related plant species [27]. Thus, for example, for growing *Hyssopus officinalis*, an optimal nutrient medium for the induction of callusogenesis was a medium with indolyl-3-butyric acid—1 mg/L, and a medium with a-naphthylacetic acid—0.5 mg/L, kinetin—0.1 mg/L and 6-benzylaminopurine—1 mg/L, and for the cultivation of plants, H. officinalis in vitro—a medium containing indolyl-3-acetic acid IAA (2 mg/L), and kinetin (0.2 mg/L) [27].

The experimental results demonstrate that the WPM medium may be more suitable for the effective multiplication of *H. cretaceous* shoots *in vitro*. A reduced concentration of nitrogen-containing salts may remain very important for this plant species (patent by Kritskaya and colleagues (2015)).

### 4.2.3. Matthiola Fragrans

The results obtained on the effect of BAP on the multiplication of *M. fragrans* differ from the information found for *M. incana* [43]. Visually, when using *m*T, the leaves of regenerating plants increased, and with KT, the shoots were slightly stretched upwards. The effects of using KT are confirmed by the literature information on the example of *M. incana* [44].

When using *m*T, the multiplication coefficient did not fall below $1.95 \pm 0.22$ per plant, which is the highest minimum value compared to those for other phytohormones.

Based on a comparative analysis of the average values of the multiplication coefficient of *M. fragrans* shoots, the use of *m*T and KT turns out to be more effective compared to the use of BAP.

The analysis of variance showed interesting data (Appendix A Table A3). Based on the results of the analysis, the mineral base B5 has a significant effect on the "work" of growth regulators. When using it, BAP and *m*T have relatively the same strength of action at all the concentrations studied. KT behaves similarly at concentrations of 1.5 and 2.0 mg/L. Of the studied factors affecting the multiplication of *M. fragrans* shoots, the type of growth regulator is the most significant. The effects of *m*T, BAP, and KT at different concentrations and different mineral bases turned out to be very different. This is especially noticeable in the graphs describing the behavior of BAP relative to other growth regulators (Figure 5). The concentration of phytohormone also affects the degree of multiplication. Based on the graphs in Figure 5, the effectiveness of using a particular hormone varies greatly from the chosen concentration.

The effectiveness of the use of *m*T on BAP was shown for many valuable and threatened species of plants in the plant world, such as *Spathifphyllum floribundum* [45], *Uniola paniculate* [46], species and varieties of *Pelargonium* [47], and *Musa* [48], *Aloe polyphylla* [12], *Actinidia chinensis*, *Coccoloba uvifera* [49], *Beta vulgaris* [50], *Cannabis sativa* [51], *Manihot esculenta* [52], *Sesamum indicum* [53], and species and varieties of *Prunus* [54].

Based on the results of statistical analysis, the use of *m*T and KT is more effective for stimulating the multiplication of *M. fragrans* in vitro compared to BAP. The MS environment is less suitable for this task compared to B5 and WPM (Figure 4). Low concentrations of phytohormones (0.5 and 1.0 mg/L) stimulate the multiplication of *M. fragrans* in vitro more effectively than elevated ones.

Different plant growth regulators belonging to the same class of phytohormones have different physiological activity for different plants. Thus, in our study, three different species from three families (*Fabaceae, Lamiaceae, Brassicaceae*) were studied. As can be seen in Figure 5 and Tables 2–4, the influence of the same growth regulator has a different effect for different species. For example, on medium B5 under the action of *m*T at a concentration of 1 mg/L, the reproduction coefficient for *H. grandiflorum* is 2.9, for *H. cretaceus*—2.05, and for *M. fragrans*—1.95. Thus, there is no universal nutrient medium for the most rare and endangered plant species, and only thanks to the mathematical planning of nutrient media, it is possible to obtain an effective scheme for the selection of a mineral base, such as a growth regulator and its concentration.

## 5. Conclusions

The method of seed sterilization used during the study made it possible to achieve high sterility rates for all three objects of study. A mixture of ethanol and hydrogen peroxide effectively cleanses the outer covers of diaspores and firing in a flame allows for additional removal of hard-seeding. In the case of sterilization of *M. fragrans* seeds, it may be necessary to reduce the exposure time in sterilizing agents to further increase sterility.

According to the results of laboratory experiments and statistical analyses, as part of a study on the selection of nutrient media and effective concentrations of phytohormones at the stage of multiplication of shoots for *H. grandiflorum*, it is recommended to use *m*T at concentrations of 0.5–2.0 mg/L on a mineral base B5; the use of BAP proved detrimental to the multiplication of *H. cretaceus* shoots *in vitro*, while the use of low (0.5–0.1 mg/L) concentra-

tions of mT and KT on a B5 and WPM medium proved effective; also, low concentrations of *m*T and KT together with nutrient media B5 and WPM are more effective for microcloning of *M. fragrans* in vitro in comparison with BAP, as well as MS nutrient medium.

For the studied rare and endangered plant species of the Rostov region that have needs for specific substrates, the MS nutrient medium rich in ammonium nitrogen adversely affects the growth and development of regenerating plants *in vitro*. A reduced concentration of ammonium in the medium or its complete absence can play a big role in the success of the selected species in vitro animation.

**Author Contributions:** Conceptualization, V.A.C., S.D.B. and O.Y.E.; methodology, S.D.B. and V.V.S.; validation, B.L.K., P.A.D. and T.V.V.; formal analysis, I.V.K.; data curation, S.D.B.; writing—original draft preparation, V.A.C. and S.D.B.; writing—review and editing, V.A.C., A.A.B. and V.D.R.; supervision, V.A.C. and T.V.V.; project administration, V.A.C. All authors have read and agreed to the published version of the manuscript.

**Funding:** The research was financially supported by the Ministry of Science and Higher Education of the Russian Federation within the framework of the state task in the field of scientific activity (no. 0852-2020-0029).

**Institutional Review Board Statement:** Not applicable.

**Informed Consent Statement:** Not applicable.

**Data Availability Statement:** Not applicable.

**Acknowledgments:** Using the equipment of the laboratory of Cellular and Genomic Technologies of Plants of the Botanical Garden of the SFedU, center of "Biotechnology, Biomedicine and Environmental Monitoring", and the center of "High Technologies".

**Conflicts of Interest:** The authors declare no conflict of interest.

## Appendix A

**Table A1.** Results of multivariate analysis of variance values of *H. grandiflorum* shoot multiplication coefficient *in vitro*.

| Effect | One-Dimensional Significance Criterion for the Number of Shoots Per Plant, pcs. (Data Table 2) Sigma-Limited Parametrization Decomposition of the Hypothesis | | | | |
| --- | --- | --- | --- | --- | --- |
| | SS | Degrees | MS | F | *p* |
| Medium type | 37.580 * | 2 * | 18.790 * | 42.277 * | 0 * |
| Hormone | 2.327 | 2 * | 1.163 | 2.617 | 0.073570 |
| Hormone concentration, mg/L | 18.733 * | 4 * | 4.683 * | 10.538 * | 0 * |
| Medium type * Hormone | 48.953 * | 4 * | 12.238 * | 27.536 * | 0 * |
| Medium type * Hormone concentration, mg/L | 59.453 * | 8 * | 7.432 * | 16.721 * | 0 * |
| Hormone * Hormone concentration, mg/L | 9.240 * | 8 * | 1.155 * | 2.599 * | 0.008249 * |
| Medium type * Hormone * Hormone concentration, mg/L | 26.713 * | 16 * | 1.670 * | 3.757 * | 0.000001 * |
| Error | 380.000 | 855 | 0.444 | | |

* The values with reliability are underlined. *p* = 0.05.

**Table A2.** Results of multivariate analysis of variance values of *H. cretaceus* shoot multiplication coefficient *in vitro*.

| Effect | One-Dimensional Significance Criterion for the Number of Shoots Per Plant. pcs. (Data Table 3) Sigma-Limited Parametrization Decomposition of the Hypothesis | | | | |
|---|---|---|---|---|---|
| | SS | Degrees | MS | F | *p* |
| Medium type | 43.469 * | 2 * | 21.734 * | 19.254 * | 0 * |
| Hormone | 18.729 * | 2 * | 9.364 * | 8.296 * | 0.000270 * |
| Hormone concentration, mg/L | 86.196 * | 4 * | 21.549 * | 19.090 * | 0 * |
| Medium type * Hormone | 90.884 * | 4 * | 22.721 * | 20.128 * | 0 * |
| Medium type * Hormone concentration, mg/L | 61.231 * | 8 * | 7.654 * | 6.780 * | 0 * |
| Hormone * Hormone concentration, mg/L | 30.338 * | 8 * | 3.792 * | 3.359 * | 0.000844 * |
| Medium type * Hormone * Hormone concentration, mg/L | 114.816 * | 16 * | 7.176 * | 6.357 * | 0 * |
| Error | 965.150 | 855 | 1.129 | | |

* The values with reliability are underlined. *p* = 0.05.

**Table A3.** Results of multivariate analysis of variance values of *M. fragrans* shoot multiplication coefficient *in vitro*.

| Effect | One-Dimensional Significance Criterion for the Number of Shoots Per Plant, pcs. (Data Table 4) Sigma-Limited Parametrization Decomposition of the Hypothesis | | | | |
|---|---|---|---|---|---|
| | SS | Degrees | MS | F | *p* |
| Medium type | 64.642 * | 2 * | 32.321 * | 22.327 * | 0 * |
| Hormone | 88.169 * | 2 * | 44.084 * | 30.453 * | 0 * |
| Hormone concentration, mg/L | 31.473 * | 4 * | 7.868 * | 5.435 * | 0.000253 * |
| Medium type * Hormone | 60.551 * | 4 * | 15.138 * | 10.457 * | 0 * |
| Medium type * Hormone concentration, mg/L | 26.180 * | 8 * | 3.272 * | 2.261 * | 0.021535 * |
| Hormone * Hormone concentration, mg/L | 46.553 * | 8 * | 5.819 * | 4.020 | 0.000106 * |
| Medium type * Hormone * Hormone concentration, mg/L | 81.993 * | 16 * | 5.125 * | 3.540 | 0.000003 * |
| Error | 1237.700 | 855 | 1.448 | | |

* The values with reliability are underlined. *p* = 0.05.

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
