# Peer review of "Investigation of Growth Factors and Mathematical Modeling of Nutrient Media for the Shoots Multiplication In Vitro of Rare Plants of the Rostov Region"

_horticulturae, doi:10.3390/horticulturae9010060_

Round 1

Reviewer 1 Report

This research provides critical information on the most successful media/hormone combinations for the regeneration of three different species that are on the endangered list.

The research assessed 3 different media types (MS, B5 and WPM) in combination with three different growth regulators (BAP, mT and KIN) at 5 different concentrations (0 - 2 mg/L)

Need to clarify that it is the shoot count that is being used to assess the success of the media combinations.  

Clarify the replication of the experiment.  Is is 3 x 20 seeds?

Figure 1. Correct the y-axis to report from 0-100.  Include the standard errors or std dev that were mentioned in the text.

There is no information on what the +/- values are.  Are they Std errors or std dev.

Analysis of results.  The first level of analysis would be to assess a single growth regulator at a time with the control for the media. e.g MS v mT (0 - 2). Use Least Significant difference to show what treatments were different.

Then look at differences between the growth regulators on the same media, e.g. did mT perform better than BAP?

Then the 3 way ANOVA to show that there were differences between the media, growth regulators and concentrations.

The discussion has all of the descriptive results that should be moved to the results section.

Tables 5-7 could be moved to a supplementary.

Consider combining Fig 2-4, so that the legend does not have to be repeated with each one.  Consider labelling the axis with the media names and the concentration levels so that you don't need to refer to the legend.  There was an issue with text overlapping in the figure on my print out.  Explain what the error bars are, they all appear to be the same size?

Results:  The statements on the results do not always match what is presented in the tables and figures. e.g. Authors say that for H.grandiflorum the highest number of shoots were observed on WPM-mT1.5mg/L (2 +- 0.21), but it is actually MS-BAP1mg/L (2.35 +- 0.26).

While there may not be significantly difference between results at some stages you can report that there is a trend for... e.g. lower shoot production than the control with growth hormone x.

Discussion: This is where you report on the previously published literature and how it compares with your results and why there may be differences.  A lot of the time the authors say that 'as indicated in the literature' but not say what was in the literature.  This applies to all three species reported.

Conclusion: The conclusions do not match the tables and figures.

Author Response

Many thanks to the distinguished reviewer for such a careful reading of our manuscript and providing a detailed review.

Need to clarify that it is the shoot count that is being used to assess the success of the media combinations.

Thank you so much for your comment. We have added clarifying information.

Clarify the replication of the experiment.  Is is 3 x 20 seeds?

Thank you so much for your comment. We have added clarifying information.

Figure 1. Correct the y-axis to report from 0-100.  Include the standard errors or std dev that were mentioned in the text. We have added clarifying information

Thank you so much for your comment. We have updated the Figure 1 according to your recommendations.

There is no information on what the +/- values are.  Are they Std errors or std dev.

Thank you so much for your comment. This information about mean and standard error

Analysis of results.  The first level of analysis would be to assess a single growth regulator at a time with the control for the media. e.g MS v mT (0 - 2). Use Least Significant difference to show what treatments were different.

Then look at differences between the growth regulators on the same media, e.g. did mT perform better than BAP?

Then the 3 way ANOVA to show that there were differences between the media, growth regulators and concentrations.

The discussion has all of the descriptive results that should be moved to the results section.

Tables 5-7 could be moved to a supplementary.

Thank you so much for your comment. We took your comment into account and made the necessary edits.

Consider combining Fig 2-4, so that the legend does not have to be repeated with each one.  Consider labelling the axis with the media names and the concentration levels so that you don't need to refer to the legend.  There was an issue with text overlapping in the figure on my print out.  Explain what the error bars are, they all appear to be the same size?

Thank you so much for your comment. We combined the figures into one. We made a common legend. And corrected the inscriptions. The same st. er. on the graph is due to the fact that the feature is discrete and has a low variation (from 1 to 3).

Results:  The statements on the results do not always match what is presented in the tables and figures. e.g. Authors say that for H.grandiflorum the highest number of shoots were observed on WPM-mT1.5mg/L (2 +- 0.21), but it is actually MS-BAP1mg/L (2.35 +- 0.26).

Thank you so much for your comment. We have corrected the inaccuracies. In some cases, when we talk about the maximum, we mean a specific medium and a specific growth regulator.

While there may not be significantly difference between results at some stages you can report that there is a trend for... e.g. lower shoot production than the control with growth hormone x.

Thank you so much for your comment. Added a part to the results.

Discussion: This is where you report on the previously published literature and how it compares with your results and why there may be differences.  A lot of the time the authors say that 'as indicated in the literature' but not say what was in the literature.  This applies to all three species reported.

Thank you so much for your comment. Added a part to the discussion.

Conclusion: The conclusions do not match the tables and figures.

Thank you so much for your comment. We have put the conclusion in the proper form. Now it is consistent with the drawing and tables.

Reviewer 2 Report

This paper studied the method of seed sterilization and shoots multiplication of rare and endangered plant species of the Rostov region. The authors found that the MS nutrient medium rich in ammonium nitrogen adversely affects the growth and development of regenerating plants in vitro. A reduced concentration of ammonium in the medium or its complete absence can play a big role in the success of the selected species in vitro animation. The results lay a foundation for establishing an effective regeneration system of these endangered plants.

Some suggestions are as follows,

1 It is suggested that the authors supplement the pictures of three plant’s shoots cultured in different media.

2 For Fig1, it is suggested to supplement the unit of vertical axis and standard deviation of data.

3 For table 5, 6 and 7, it is suggested to change the 0.000,00 to 0 or a specific value.

4 For Fig2, 3 and 4, it is suggested to supplement the unit of vertical axis.

5 For Results, the author's description of the analysis of the test results is too little. The author put a lot of data analysis results into ‘Discussion’, which is not appropriate. It is suggested that the author write the description of the result analysis and the discussion separately.

6 For discussion, it is suggested that the author compare and analyze the results obtained in this paper with those obtained by others.

7 Different plant growth regulators have different concentrations of physiological activity. The author uses some unified concentrations to measure effect of different plant growth regulators. In different plant species, the culture effect must be different. However, the author did not compare the effects of the same plant growth regulator and the same concentration on different species. It is suggested that the author should add the analysis of differences between species.

8 It is suggested to change the KN to KT.

Author Response

Many thanks to the distinguished reviewer for such a careful reading of our manuscript and providing a detailed review.

1 It is suggested that the authors supplement the pictures of three plant’s shoots cultured in different media.

Thank you so much for your comment. We have included several figures in the manuscript.

2 For Fig1, it is suggested to supplement the unit of vertical axis and standard deviation of data.

Thank you so much for your comment. We have updated the Figure 1 according to your recommendations.

3 For table 5, 6 and 7, it is suggested to change the 0.000,00 to 0 or a specific value.

Thank you so much for your comment. We changed.

4 For Fig2, 3 and 4, it is suggested to supplement the unit of vertical axis.

Thank you so much for your comment. We have updated.

5 For Results, the author's description of the analysis of the test results is too little. The author put a lot of data analysis results into ‘Discussion’, which is not appropriate. It is suggested that the author write the description of the result analysis and the discussion separately.

Thank you so much for your comment. We took your comment into account and made the necessary edits.

6 For discussion, it is suggested that the author compare and analyze the results obtained in this paper with those obtained by others.

Thank you so much for your comment. We compared our results with the results of other authors on the plants that were studied.

7 Different plant growth regulators have different concentrations of physiological activity. The author uses some unified concentrations to measure effect of different plant growth regulators. In different plant species, the culture effect must be different. However, the author did not compare the effects of the same plant growth regulator and the same concentration on different species. It is suggested that the author should add the analysis of differences between species.

Thank you so much for your comment. We added a description at the end of the discussion.

8 It is suggested to change the KN to KT.

Thank you so much for your comment. We changed.

Round 2

Reviewer 1 Report

No comments.

Author Response

Many thanks

Reviewer 2 Report

(1) It is suggested to change the mg/l to mg/L. The L should be capital letter. (2) It is suggested to supplement the Bars(scale) in the Figure 3, 4 and 5. (3) For *The values with reliability are underlined. P = 0.05., the P should be italicized.

Author Response

Many thanks to the distinguished reviewer for such a careful reading of our manuscript and providing a detailed review.

(1) It is suggested to change the mg/l to mg/L. The L should be capital letter.

Thank you so much for your comment. We have replaced everything in the text, including the drawings. Everywhere they wrote mg/L.

(2) It is suggested to supplement the Bars(scale) in the Figure 3, 4 and 5.

Thank you so much for your comment. Unfortunately, at the moment it is not possible to take new photos with the scale. Plants are already at the next stage of experiments. And it will not be possible to attach an artificial scale to these photos, because the pictures were taken on the phone from different focal lengths.

(3) For *The values with reliability are underlined. P = 0.05., the P should be italicized.

Thank you so much for your comment. We have corrected everything according to your comments.